# Learning reliable rules by re-generating deep features

## Abstract

Improving the interpretability and reliability of deep learning models is essential for advancing machine learning applications, though it remains a significant challenge. One promising approach is the integration of logical reasoning into deep learning systems. Previous works have demonstrated that SATNet, a differentiable MaxSAT solver, can learn interpretable and reliable rules from input-output examples in puzzle domains. In this work, we propose *Visual SATNet* (Vi-SATNet), an extended version of SATNet capable of learning logical reasoning rules in more general and complex domains, such as the feature space of real-life images. We find that, given a pre-trained deep convolutional neural network (CNN) architecture, a Vi-SATNet layer can be integrated and trained efficiently to learn a set of reasoning rules on the deep features, guiding the classifier's decision. Vi-SATNets are trained to perform feature re-generation tasks for a given image dataset, where the re-generated features maintain high accuracy when used for image classification, proving their quality. In our experiment on the Imagenette dataset with a pre-trained VGG19 model, masking out 10% to 80% of the features results in classification accuracy ranging from 98.50% to 93.92% with Vi-SATNet re-generation, compared to 97.07% to 9.83% without re-generation. Furthermore, we introduce a visualization method to illustrate the rules learned by Vi-SATNets, thereby enhancing the interpretability of the pre-trained CNN model.

## 1 Introduction

> What I cannot create, I do not understand.
>
> *– Richard Feynman*

Deep convolutional neural networks (CNNs) have been the foundation of numerous vision tasks ranging from image classification (Krizhevsky et al., 2012), object localization (Zhou et al., 2016), image segmentation (He et al., 2017), face recognition (Parkhi et al., 2015), medical image processing (Ronneberger et al., 2015), autonomous driving (Maqueda et al., 2018) among many others. Although remarkably effective, the deep features learned through CNNs are still poorly understood. Many existing works highlight corresponding regions of input via saliency methods (e.g., visualization, maximum activation, attribution methods) (Simonyan et al., 2014; Sundararajan et al., 2017; Shrikumar et al., 2017) or perturbing inputs (Fong et al., 2019; Ivanovs et al., 2021), which appear to be visually convincing but can be misleading (Adebayo et al., 2018; Laugel et al., 2019). Inspired by the fact that neural networks fundamentally learn distributed representations (Hinton, 1986), we believe a more meaningful way of interpreting deep features is to understand their *interactions* rather than visualizing or analyzing individual features.

Recent works Wang et al. (2019); Topan et al. (2021); Lim et al. (2022); Li et al. (2023) show that SATNet, neural networks with a special layer for differentiable MaxSAT solving, can learn reliable logical rules in the form of weighted equalities. SATNet has proven to be successful in solving small logical puzzles like Sudoku but suffers from training instability when trained end-to-end with other neural network layers Chang et al. (2020). In this work, we propose a *decomposable* and *interpretable* layer based on SATNet, which is a drop-in component that can be directly integrated into pre-trained CNNs and requires no re-training or finetuning.

Figure 1: An Overview of the Vi-SATNet Architecture. A feature map with $K$ channels is shown as a stack of 2-dimensional grids. The coloured grids shows the formation of a *feature vector*. The coloured along with the grey grids indicate input (known) feature values; the white grids indicate masked feature values; the black grids indicate feature values completed by re-generation. The feature extractor is taken out-of-box from pre-trained CNNs.

In summary, we make the following contributions:

- We introduce ***Visual SATNet*** (Vi-SATNet), a generalized formulation of SATNet that is capable of learning logical rules among deep features (an overview of the framework can be found in Figure 1).

- We propose a novel interpretation method of deep features by integrating drop-in Vi-SATNet models into CNN architectures.

- We train and evaluate Vi-SATNet models on deep feature re-generation, demonstrating the high quality of the re-generated features by testing them on image classification tasks.

- We present a visualization technique to illustrate the rules learned by Vi-SATNet models.

## 2 RELATED WORK

**Interpreting deep features.** There has been significant research effort in understanding deep features of convolutional neural networks, most of which are post-hoc explanation methods using techniques such as activation maximization (Erhan et al., 2009; Yosinski et al., 2015; Yoshimura et al., 2021), image perturbation (Fong et al., 2019; Ivanovs et al., 2021) and saliency maps (Simonyan et al., 2014; Smilkov et al., 2017; Sundararajan et al., 2017). These approaches focus on explaining individual features and can be quite problematic (Adebayo et al., 2018; Laugel et al., 2019). Recent works (Chen et al., 2019; Ma et al., 2023) introduce explanations with prototypical concepts, which are reference images that share similar patterns. This requires a set of carefully chosen references and some manual inspection is needed to validate whether the test image is indeed similar to reference images. In contrast, our work aims to learn logical rules among deep features and further automatically identify essential features with learned rules and recover them when missing.

**Learning and reasoning with SATNet.** Wang et al. (2019) propose SATNet, a differentiable layer approximating MaxSAT solving through the semi-definite programming (SDP) relaxation, and show that it can learn implicit rules and solve logical puzzles like Sudoku. Chang et al. (2020) find a label-leakage issue in the original SATNet and point out the symbol grounding challenge, which is partly addressed by a staged training (Topan et al., 2021). Lim et al. (2022) introduce regularizations like symmetric constraints, improving the learning efficiency of SATNet when the targeted domain shares intrinsic symmetries. More recently, Li et al. (2023) proposes a reliable interpretation of rules implicitly learned in SATNet and shows they are verifiably correct for small logical puzzles. All these works focus on learning and solving logical puzzles and assume the input to SATNet is properly binarized based on domain knowledge. Our work generalizes SATNet without the need to design any binary features based on a simple and practical observation — deep features from CNNs are already good relaxations and can be directly used as relaxed inputs to SATNet. This simple observation makes SATNet broadly applicable to state-of-the-art CNNs and beyond.

## 3 VI-SATNET

Prior work has shown the reliability and interpretability of SATNet (Li et al., 2023; Wang et al., 2019), however, its applications are limited to learning and solving logic puzzles such as Sudoku. In this section, we propose *Visual SATNet* (Vi-SATNet), a generalized architecture that builds upon SATNet and is capable of learning logical rules directly from the feature embedding space of images.

### 3.1 PROBLEM FORMULATION

Recall that the objective of a SATNet layer is to learn a set of rules such that when applied, unknown variables in a given incomplete puzzle can be solved. Similarly, the objective of a Vi-SATNet model is to learn a set of rules such that when applied, missing feature values (extracted from an image) in a given incomplete feature map can be re-generated. In addition, we aim to use the learned rules to explain how the features in a given class of images interact with each other.

Formally, let $\mathcal{M}$ denotes a convolutional neural network (CNN) pre-trained on an image dataset $\mathcal{D} = \{(\boldsymbol{x}_i, y_i)\}_{i=1}^N$, where $\mathcal{M}$ consists of a feature extractor $\mathcal{F}$, the set of convolutional layers and pooling layers, and a feature classifier $\mathcal{C}$, the set of fully connected layers (Lecun et al., 1998). Given an image $\boldsymbol{x}$, $\mathcal{F}(\boldsymbol{x})$ generates a feature map $F \in \mathbb{R}^{H \times W \times K}$, where $H$ and $W$ are the height and width of the feature map, $K$ is the number of channels. Let $f^{(h,w,k)} \in \mathbb{R}$ denote the feature map values at position $(h, w)$ in the $k$-th channel of $F$.

Consider fixed mappings $\mathcal{F}$ and $\mathcal{C}$ from a pre-trained $\mathcal{M}$; let $n = H \times W$ be the number of *feature vectors*, where a feature vector $v_i$ is built by taking the feature values at *the same positions* in each of the channels:

$$v_i = (f^{(\lfloor \frac{i}{w} \rfloor, i \bmod w, 1)}, f^{(\lfloor \frac{i}{w} \rfloor, i \bmod w, 2)}, \ldots, f^{(\lfloor \frac{i}{w} \rfloor, i \bmod w, K)}) \in \mathbb{R}^K, i \in \{1, \ldots, n\}, \quad (1)$$

where $(\lfloor \frac{i}{w} \rfloor, i \bmod w)$ marks the position on the 2-dimensional plane of a feature map channel.

Now, define $\mathcal{I} \subset \{1, \ldots, n\}$ to be the index set for known feature vectors and $\mathcal{O} \equiv \{1, \ldots, n\} \setminus I$ to be the index set for missing feature vectors. The **feature re-generation** task is to generate vectors $\{v_o\}_{o \in \mathcal{O}}$ such that the distance between the generated feature vectors $\{\hat{v_o}\}_{o \in \mathcal{O}}$ and the ground truth feature vectors $\{v_o^{gt}\}_{o \in \mathcal{O}}$ is minimized by some distance metric $\mathbf{d}$.

### 3.2 VI-SATNET AS A FEATURE RE-GENERATION MODEL

As illustrated in Figure 1, a Vi-SATNet model directly takes as input the feature vectors $V = [v_1, \ldots, v_n]$ and a binary mask that indicates the missing indices. Following Wang et al. (2019), we then apply the mixing method to obtain the optimized set of output feature vectors (Wang et al., 2017). In this context, a feature vector $v_i$ can be viewed as the relaxation of a binary variable $z_i$ that encodes some abstract status of the corresponding local region of the input image during the inference process. This interpretation is discussed in more detail in Section 3.4.

During a forward pass of Vi-SATNet, a fraction of the input feature vectors are masked, and the randomly initialized weight matrix $S$ is used to optimize the generation of the unknown feature vectors $\hat{V}_{\mathcal{O}}$. We use cosine similarity defined in Equation 2 Nguyen & Bai (2011) to quantitatively assess the similarity between the two vectors $\hat{v_o}$ and $v_o^{gt}$:

$$\mathbf{d}(v_o, v_o^{gt}) = \frac{v_o \cdot v_o^{gt}}{\|v_o\| \|v_o^{gt}\|}. \quad (2)$$

During a backward pass of Vi-SATNet, the loss is propagated to update the weight matrix $S$ to best fit the input-output pairs provided. Upon convergence, $S$ encodes a set of learned rules. The implementation of Vi-SATNet is generally adapted from the original implementation of a SATNet layer, and the detailed algorithms can be found in Appendix A. Note that in the formulation of SATNet, the relaxed vectors are assumed to be unit vectors. This assumption can easily be met by normalizing the feature vectors, and we show that normalization does not cause information loss in Section 4.2.

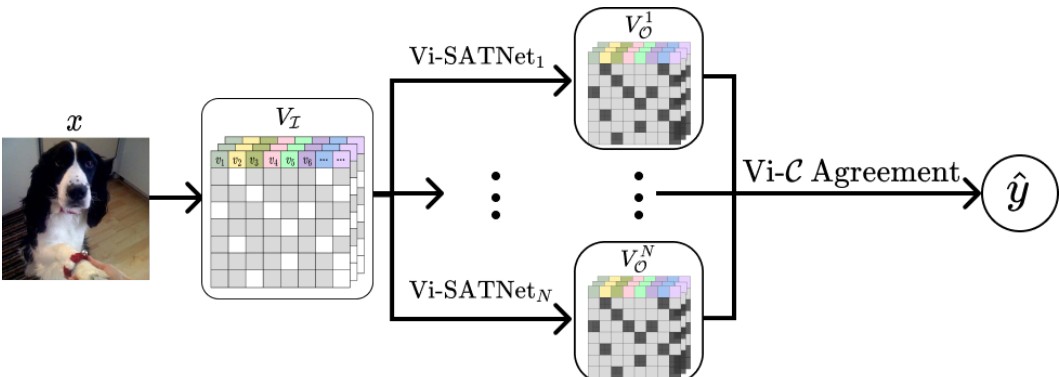

Figure 2: Vi-SATNet integrated in CNN for classification inference. Feature values obtained via applying $\mathcal{F}(\mathbf{x})$ is re-shaped into feature vectors, where a mask is applied to cover a set of vectors. The input feature vectors $V_{\mathcal{I}}$ is then passed to all $N$ Vi-SATNets to perform feature vector re-generation. Each feature map completed by re-generation is send to the pre-trained classifier $\mathcal{C}$ to perform Vi-$\mathcal{C}$ agreement to obtain the final prediction label.

### 3.3 EVALUATION OF THE QUALITY OF RE-GENERATED FEATURES

**Cosine Similarity.** A trivial method to evaluate the quality of re-generated feature vectors is to compute the cosine similarity as stated in Equation 2. However, this evaluation metric is not able to assess the degree of usefulness of the re-generated feature vectors since a single similarity score is not expressive enough to demonstrate whether the key information encoded in the ground truth feature vectors is well-captured by the re-generated ones or not.

**Integrating Vi-SATNets into CNN models.** To quantitatively evaluate the quality of re-generated feature vectors, we propose to integrate trained Vi-SATNets into the pre-trained CNN model and use the pre-trained classifier $\mathcal{C}$ to perform image classification based on the re-generated feature vectors. The evaluation metric is simply the classification accuracy of $\mathcal{C}$ when applied to the re-generated features.

Intuitively, images from the same labelled class should share the same set of reasoning rules that capture the interactions between the feature vectors. To reduce the complexity of the learned rules, we propose training separate Vi-SATNets for each class. One of the main advantages of this approach is that each Vi-SATNet can be trained independently, which ensures that no re-training of other pre-existing Vi-SATNets is needed when images of new classes are introduced to the dataset, thus minimizing any unnecessary computational costs.

---

**Algorithm 1:** VI-$\mathcal{C}$ AGREEMENT

**Input:** Input feature $V_{\mathcal{I}}$ from $\mathcal{M}$, Label $l \in \{1, ..., N\}$, Set of Vi-SATNets $S = s_l \in \{s_1, ..., s_N\}$
**Output:** Predicted label $\hat{y}$

1  **Function** *Vi-C_Agreement($V_{\mathcal{I}}$, S)*
2     **for** $s_l \in S$ **do**
3        $V_{\mathcal{O}}^l := s_l(V_{\mathcal{I}})$
4        Obtain $\hat{y}_l$ from $\mathcal{C}(V_{\mathcal{O}}^l)$
5        **if** $l == \hat{y}_l$ **then**
6           Record prediction $\hat{y}$ as $l$

---

An overview of the evaluation framework is given in Figure 2. Given a CNN model $\mathcal{M}$, we train and attach $\{\text{Vi-SATNet}\}_{l=1}^N$ where $N = |\mathcal{D}|$ is the number of classes in the dataset. Each Vi-SATNet$_l$ is trained solely on images from label $l$. For each input image, $V_{\mathcal{I}}$ is obtained by applying a random mask, which is then passed into all $N$ Vi-SATNets to perform feature re-generation. During this process, each Vi-SATNet tries to maximally recover the missing information from the given incomplete feature vectors according to the rules stored in the weight matrix $S_l$. To compute the classification prediction, we introduce the ***"Vi-C Agreement"*** criteria shown in Algorithm 1. If $S_l$ successfully

recovers all key information via re-generating high quality $V_{\mathcal{O}}$, then $\mathcal{C}$ is expected to predict label $l$ with high confidence. In this case, we say that there is an agreement between Vi-SATNet and $\mathcal{C}$, hence, $l$ is recorded as the predicted label for the input image.

## 3.4 INTERPRETATION OF THE LEARNED RULES

As mentioned in Section 3.2, each of the feature vectors conceptually corresponds to a binary variable that represents an abstract state of this feature vector. When a Vi-SATNet model converges, the weight matrix $S$ encodes the logical relations between each pair of the feature vectors. During the inference phase, the unknown feature vectors $V_{\mathcal{O}}$ are generated such that the entire set of feature vectors $V_{\mathcal{I} \cup \mathcal{O}}$ optimally respects the rules encoded in $S$. Although we can extract explicit rules from a Vi-SATNet using the method introduced in Li et al. (2023), it is not clear what should be considered as the "ground truth rules" in the context of describing feature relations. Hence, we cannot perform formal verification on the set of rules learned by Vi-SATNet. It is worth noting that this is not a technical limitation but the lack of formal specifications of visual objects — e.g., we do not have a set of ground-truth logical rules specifying what a cat is. So instead, we propose to *visually illustrate* the learned rules.

### 3.4.1 FINDING THE MINIMAL SIGNIFICANT FEATURE SET

Given an image $\boldsymbol{x}$ from a class $y$, a Vi-SATNet model trained for label $y$ is capable of re-generating any missing feature vector. We can thus demonstrate the meaning of the learned rules by studying the inter-dependency between the feature vectors during the re-generation process.

**Definition 1** (Significant Feature Set). *Given a target missing feature vector $v_t$ in an image $\boldsymbol{x}$ with label $y$, a **significant feature set** is a set of feature vectors which, when passed into Vi-SATNet$_y$, can re-generate $\hat{v}_t$ that is similar to the ground truth vector $v_t$. Let $\mathcal{S}_f \subset \{1, \ldots, n\}$ be the index set of the significant feature vectors, then information from all other feature vectors with indices $\{1, \ldots, n\} \setminus (\{t\} \cup \mathcal{S}_f)$ are removed by setting them to uniform unit vectors before passed into Vi-SATNet$_y$, where a uniform unit vector $v \in \mathbb{R}^k$ is defined as $v = \left[ \frac{1}{\sqrt{k}}, \frac{1}{\sqrt{k}}, \ldots, \frac{1}{\sqrt{k}} \right]^{\top}$.*

There are exponentially many significant feature set that can be found for a given target feature vector. In the following definitions, we introduce a measure of confidence to characterize the space of these sets.

**Definition 2** (Reference Similarity Score). *Given a target missing feature vector $v_t$ in an image $\boldsymbol{x}$ with label $y$, the **reference similarity score** $\boldsymbol{s_{ref}}$ is defined as the cosine similarity between $\hat{v}_t$ and $v_t$ given all feature vectors excluding $v_t$ itself:*

$$\boldsymbol{s_{ref}} = \mathbf{d}(\hat{v}_t, v_t \mid \{v_i\}_{i=1}^n \setminus \{v_t\}). \tag{3}$$

**Definition 3** (Significant Feature Set of $\alpha$ confidence). *We say that a significant feature set is **of confidence $\alpha$** when*

$$\mathbf{d}(\hat{v}_t, v_t \mid \{v_i\}_{i \in \mathcal{S}_f}) \geq \alpha \cdot \boldsymbol{s_{ref}}, \quad \alpha \in [0, 1]. \tag{4}$$

It is not hard to see that there exists a significant set of confidence $\alpha$ for *any* value of $\alpha$: simply taking $\mathcal{S}_f = \{v_i\}_{i=1}^n \setminus \{v_t\}$. Our goal is to find a minimal such set, so that the remaining feature vectors in the set have the most influence on the target feature vector.

A minimal significant feature set can be found by enumerating over all significant feature sets fixing an $\alpha$ value, however, this approach results in an exponential search space. Inspired by the delta debugging technique used in automated bug detection in programming (Zeller, 1999), we introduce a search algorithm that efficiently finds an approximated minimal significant feature set (See Algorithm 2).

---

**Algorithm 2:** FINDMINIMALSIGNIFICANTFEATURESET

---

**Input:** Target index $t$, full set of valid indices $full\_set = \{i\}_{i=1}^{n} \setminus \{t\}$, threshold $\theta = \alpha \cdot \mathbf{s}_{\text{ref}}$
**Output:** Subset of features $min\_significant\_set = full\_set \setminus test\_set$

1 **Function** *FindLargestToDrop($full\_set$, $\theta$)*
2     $test\_set \leftarrow \emptyset$
3     $queue \leftarrow$ priority queue containing $full\_set$ with priority equal to its size
4     **while** *queue is not empty* **do**
5        $current\_set \leftarrow$ queue.pop() /* Get the subset with the largest size      */
6        $cos\_sim \leftarrow \mathbf{d}(\hat{v}_t, v_t \mid \{v_i\}_{i \notin current\_set})$
7        **if** $cos\_sim > \theta$ **then**
          /* Not affecting score significantly, can be dropped     */
8           $test\_set \leftarrow test\_set \cup current\_set$
9        **else**
10           split $current\_set$ by half into $left$ and $right$
11           $queue.push(left)$ with priority equal to size of $left$
12           $queue.push(right)$ with priority equal to size of $right$
13     **return** $test\_set$

---

### 3.4.2 VISUALIZING THE LEARNED RULES

Since a receptive field (all image pixels that contribute to the calculation of the feature value) can be calculated for each feature vector, the minimal significant set found by Algorithm 2 can be visually mapped onto the input image. For deep convolutional neural networks, the theoretical receptive field size can be as large as, or even larger than the input image due to the padding operation. Therefore, instead of plotting the theoretical receptive fields, we plot the *effective receptive fields* (Luo et al., 2016) of the minimal significant feature set on the input image to show the most relevant features used during re-generation of a target feature vector $v_t$.

## 4 RESULTS

In this section, we present empirical results to demonstrate the learning ability and interpretability of Vi-SATNet models. All experiments are carried out on a single 3090ti GPU and 16 CPU cores. For model training, the optimizer, learning rate, and hyperparameters are set to be the same as in Wang et al. (2019) if not otherwise stated. All models are trained for 50 epochs.

### 4.1 TRAINING VI-SATNET MODELS

All Vi-SATNet models presented in this section are trained on the feature-regeneration objective with the cosine similarity loss

$$1 - \mathbf{d}(V_{\mathcal{O}}, V_{\mathcal{O}}^{gt}), \tag{5}$$

where $\mathbf{d}$ is defined in Equation 2.

**MNIST.** A set of 10 Vi-SATNets is trained on the MNIST handwritten digits dataset (Deng, 2012) with $\mathcal{F}$ and $\mathcal{C}$ obtained from a pre-trained LeNet-5 (Lecun et al., 1998). The dimension of the feature map extracted by $\mathcal{F}$ is $H = 5, W = 5, K = 16$. All 10 Vi-SATNet are trained with the following hyperparameters: $n = H \cdot W = 25$, $m = 320$, $K = 16$, $batch\_size = 8$, $mask\_ratio = 40\%$. The loss trajectory for training and validation is shown in Figure 3a.

The plotted loss value is the average cosine similarity loss per image per re-generated feature vector, trained on digits 0 to 9. We can see that the average training loss decreased to around 0.16, which is a very low value considering the fact that cosine similarity loss ranges from 0 to 2. We observe that Vi-SATNets converge very quickly (at around epoch 30), and generalizes well to the validation set of unseen data: the average training loss almost overlaps with the average validation loss, while the individual training losses are not included on the plot but they are also very close to their respective validation losses. An interesting observation is that digit "1" is the easiest to learn (converges to the lowest loss), while digit "8" is the hardest. We anticipate that digit "8" requires a set of rules that is the most complex among all other digits.

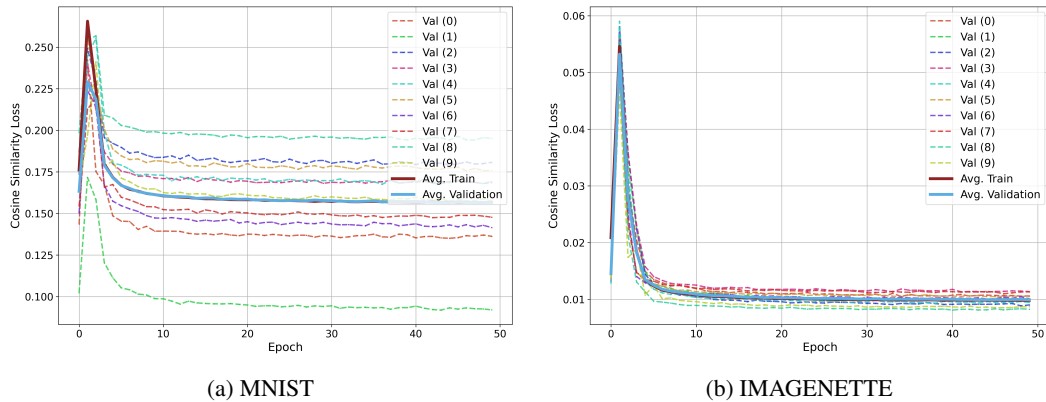

(a) MNIST                                      (b) IMAGENETTE

Figure 3: Vi-SATNet validation loss trajectory over epochs for different labels. The solid lines are the average training and validation loss respectively.

Table 1: Classification accuracy with different mask ratio on Imagenette. Mean accuracy and error bar reported on 10 runs for each mask ratio.

| Mask Ratio (%) | w/o Regeneration (%) | w Regeneration (%) |
|---|---|---|
| 10 | 97.20±0.06 | 98.48±0.01 |
| 20 | 83.50±0.09 | 98.49±0.01 |
| 30 | 46.20±0.08 | 98.49±0.02 |
| 40 | 23.38±0.05 | 98.44±0.03 |
| 50 | 11.73±0.05 | 98.32±0.03 |
| 60 | 9.85±0.01 | 98.13±0.05 |
| 70 | 9.84±0.004 | 97.45±0.05 |
| 80 | 9.83±0.001 | 94.23±0.10 |
| 90 | 9.84±0.003 | 72.87±0.18 |
| 100 | 9.84±0.005 | 6.83±0.11 |

**Imagenette.** We have also trained a set of Vi-SATNets on a 10-class subset of ImageNet (Deng et al., 2009), namely the Imagenette dataset (Howard et al., 2019), with $\mathcal{F}$ and $\mathcal{C}$ obtained from a VGG19 model pre-trained on the ImageNet dataset available in PyTorch(Simonyan & Zisserman, 2014). $\mathcal{C}$ is modified such that the prediction only targets one of the 10 classes included in Imagenette. The dimension of the feature map extracted by $\mathcal{F}$ is $H = 7, W = 7, K = 512$. All 10 Vi-SATNet are trained with the following hyperparameters: $n = H \cdot W = 49$, $m = 500$, $K = 512$, $batch\_size = 8$, $mask\_ratio = 40\%$. The loss trajectory for training and validation is shown in Figure 3b.

Again, we observe that the model can learn to re-generate high-quality feature vectors within 30 epochs, while generalizing well to unseen data. The training and validation loss drops to 0.01 at convergence. In this dataset, there is no significant difference in the complexity of individual labels.

### 4.2 IMAGE CLASSIFICATION USING RE-GENERATED FEATURES

To evaluate the quality of the re-generated feature vectors, we take the Vi-SATNet models trained on the Imagenette dataset in Section 4.1 to perform image classification based on the Vi-$\mathcal{C}$ agreement introduced in Algorithm1.

We test the re-generation performance of Vi-SATNets with mask ratios ranging from 10% to 100%, where each of the masked vectors $\{v_o\}_{o \in \mathcal{O}}$ are replaced by a *uniform vector* (see Definition1). The classification accuracy obtained by Vi-$\mathcal{C}$ agreement (with Vi-SATNets re-generation) is compared to the baseline accuracy obtained by direct classification using $\mathcal{C}$ without Vi-SATNet re-generation. The results are summarized in Table 1.

**Discussion.** We observe that Vi-SATNet demonstrates exceptional performance in the re-generation task, outperforming the baseline on all masking ratio values. It is worth highlighting that all Vi-SATNets are only trained with a mask ratio of 40%, however, we can see that its performance generalizes to all other mask ratios: even applying a mask ratio of 80% still achieves a testing accuracy of 93.92%. This shows that the models successfully capture useful reasoning rules for feature vector generation, and the more input information given, the more accurate the re-generated unknown feature vectors are; it is expected that 100% masking gives a low accuracy since in our formulation, the Vi-SATNet models assume a non-empty set of known input feature vectors.

**Remark.** The classifier $\mathcal{C}$ used in Vi-$\mathcal{C}$ agreement is taken directly from the pre-trained VGG19. It is originally trained on un-normalized deep features. However, the output feature vectors $V_{\mathcal{O}}$ re-generated by Vi-SATNet are all unit vectors by construction. We observe that *without* re-training any weight matrix inside $\mathcal{C}$, the re-generated feature vectors (which are normalized) maintain a high accuracy across different mask ratios, showing that the re-generated feature vectors accurately capture information contained in the original deep features.

### 4.3 VISUALIZING THE LEARNED RULES

Recall from Section 3.4 that a minimal significant feature set (MSF) can be found by Algorithm 2, where for a target feature vector $v_t$, an MSF of confidence $\alpha$ gives a set of most relevant feature vectors during the process of re-generating $v_t$. We present visual illustrations of the learned reasoning rules via mapping the effective receptive fields (ERFs) of the MSFs found in the following case studies; all images used in this section are from the Imagenette dataset.

**Case Study 1: Location distributions of an MSF.** By the construction of the feature vectors, neighbouring vectors contain similar information since their receptive fields (or effective receptive fields) have some overlapping regions. This pattern is found in Figure 4a and Figure 4b: when $v_t$ (highlighted in **red**) has an ERF centred at the location of an object, the MSF found are the neighbouring features that are also centred at the object of interest. Figure 4c and Figure 4d give examples of a more interesting pattern: given $v_t$, the MSF found does not only contain neighbouring feature vectors, but also includes some feature vectors located further away. In both cases, we observe that the feature vectors included in the MSF all have RFs congregating on the object of interest.

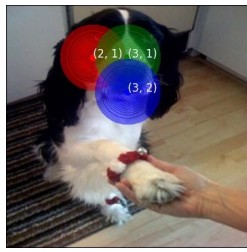 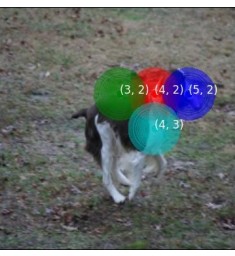 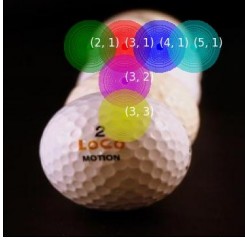 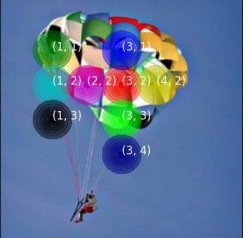

(a) Class label: English Springer, $\alpha = 0.9$

(b) Class label: English Springer, $\alpha = 0.9$

(c) Class label: golf ball, $\alpha = 1.0$

(d) Class label: parachute, $\alpha = 1.0$

Figure 4: Distribution of MSF locations. The target feature vector to be generated is always highlighted in the red colour. The other colours highlight the effective receptive fields of feature vectors in the MSF.

**Case Study 2: Foreground and Background.** It is well-expected that to generate $v_t$ centring at the object of interest in the image (in other words, the foreground), the most relevant information that should be given as input is a set of feature vectors that are also centred at some foreground regions. What about when the target is in the background? In Figure 5a and Figure 5c, we shift $v_t$ from the foreground to the background, and observe that the location pattern of the MSF changes accordingly: for the same input image, we can see that a $v_t$ in the foreground only relies on some other feature vectors in the foreground to be re-generated, while a $v_t$ in the background has an MSF of all background-centred feature vectors. This observation illustrates that the learned rules are aware of the information contained in each feature vectors, and can perform reasoning accordingly.

**Case Study 3: Effect of $\alpha$.** We further investigate how the confidence level $\alpha$ affect the MSF found for a given target feature vector. In Figure 5a and Figure 5b we can see that increasing $\alpha$ from 0.9 to 0.98 results in an additional three feature vectors added to the MSF. This behaviour is anticipated, as generating a more accurate $v_t$ requires incorporating a larger number of known feature vectors into the input.

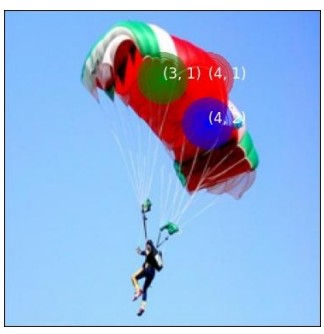 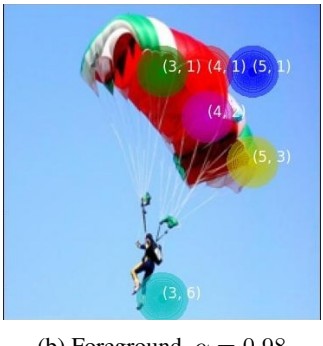 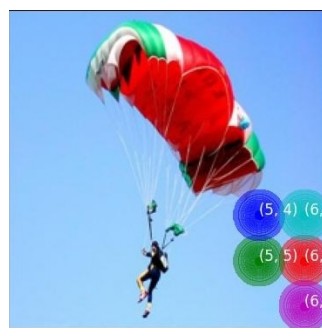

(a) Foreground, $\alpha = 0.9$      (b) Foreground, $\alpha = 0.98$      (c) Background

Figure 5: Comparison of foreground and background target vectors; Effect of increasing $\alpha$ on MSF size. Once again, the target feature vector to be generated is always highlighted in the red colour.

With the above three case studies, we can see that Vi-SATNet is capable of capturing a set of meaningful rules for each class of images using the feature vectors learned by a pre-trained CNN model. By visualizing the interactions between feature vectors during the re-generation process, the interpretability of the CNN model is enhanced, as this allows us to examine the correspondence between the learned features and better understand the reasoning process.

## 5 CONCLUSION

Achieving interpretability in deep neural networks is a critical yet challenging task. To this end, we propose Vi-SATNet, a standalone component that learns and performs reasoning on the deep feature space produced by convolutional neural networks. Empirical results on the features extracted by a pre-trained Lenet-5 model (on the MNIST dataset) and by a pre-trained VGG19 model (on the Imagenette dataset) show that the learned reasoning rules allow Vi-SATNet to re-generate missing feature vectors accurately. We further illustrate the dependencies among the feature vectors via visualizations of minimal significant feature sets, enhancing the interpretability of the deep feature space learned by the VGG19 model.

Our experiments on the shallow neural network (Lenet-5) and deep neural network (VGG19) suggest that Vi-SATNet has the potential to be generalized to any feature space learned by other convolutional neural networks, for example, ResNet (He et al., 2016). Future work should investigate a wider range of model architectures that can integrate Vi-SATNet as a drop-in reasoning layer, and extend the evaluations to larger datasets such as the ImageNet dataset (Deng et al., 2009).

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

## A  APPENDIX

Below we present the full algorithm for a Vi-SATNet model, which is adapted from the original implementation of SATNet (Wang et al., 2019).

---

**Algorithm 3:** Vi-SATNet

---
1 **Procedure** INIT()
    // Number of clauses, number of variables and auxiliary variables,
       initial weights
2   Initialize $m, n, n_{\text{aux}}, S$
3 **Procedure** FORWARD($\mathbf{V}_{\mathcal{I}}$)
    // Compute $\mathbf{V}_{\mathcal{O}}$ from $\mathbf{V}_{\mathcal{I}}$ via coordinate descent (Algorithm 4)
4   Compute $\mathbf{V}_{\mathcal{O}}$
5   **return** $\mathbf{V}_{\mathcal{O}}$
6 **Procedure** BACKWARD($\frac{\partial \ell}{\partial \mathbf{V}_{\mathcal{O}}}$)
    // Compute $\mathbf{U}$ from $\frac{\partial \ell}{\partial \mathbf{V}_{\mathcal{O}}}$ via coordinate descent (Algorithm 5)
7   Compute $\mathbf{U}$
    // Compute $\frac{\partial \ell}{\partial \mathbf{V}_{\mathcal{I}}}$ and $\frac{\partial \ell}{\partial S}$ from $\mathbf{U}$ via (Equation 12, Equation 11 in Wang
       et al. (2019))
8   Compute $\frac{\partial \ell}{\partial \mathbf{V}_{\mathcal{I}}}, \frac{\partial \ell}{\partial S}$
9   **return** $\frac{\partial \ell}{\partial \mathbf{V}_{\mathcal{I}}}$

---

---

**Algorithm 4:** Forward Pass Coordinate Descent

---
**Input:** $\mathbf{V}_{\mathcal{I}}$                        // Inputs for known variables
**Output:** $\mathbf{V}_{\mathcal{O}}$            // Final guess for output columns of $\mathbf{V}$
  // Initialize $\mathbf{v}_o$ as uniform unit vectors
1 Initialize $\mathbf{v}_o^{\text{u}}, \forall o \in \mathcal{O}$
  // Compute initial $\Omega$
2 Compute $\Omega = \mathbf{V}\mathbf{S}^{\top}$
3 **while** *not converged* **do**
4   **for** $o \in \mathcal{O}$                // For all output variables
5     **do**
6       Compute $g_o = \Omega s_o - \|s_o\|^2 v_o$ Compute $v_o = -g_o/\|g_o\|$
7       Update $\Omega = \Omega + (v_o - v_o^{\text{prev}})s_o^{\top}$

---

---

**Algorithm 5:** Backward Pass Coordinate Descent

---
**Input:** $\left\{ \frac{\partial \ell}{\partial v_o} \mid o \in \mathcal{O} \right\}$             // Gradients w.r.t. outputs
**Output:** $U_{\mathcal{O}}$
  // Compute $U_{\mathcal{O}}$ from Equation (9)
1 Compute $U_{\mathcal{O}}$
  // Initialize $U_{\mathcal{O}} = 0$ and $\Psi = 0$
2 Initialize $U_{\mathcal{O}} = 0$ and $\Psi = (U_{\mathcal{O}})S_{\mathcal{O}}^{\top} = 0$
3 **while** *not converged* **do**
4   **for** $o \in \mathcal{O}$                // For all output variables
5     **do**
6       Compute $d_{g_o} = \Psi s_o - \|s_o\|^2 u_o - \frac{\partial \ell}{\partial v_o}$
7       Compute $u_o = -P_o d_{g_o}/\|g_o\|$
8       Update $\Psi = \Psi + (u_o - u_o^{\text{prev}})s_o^{\top}$

---

