# SUPPLEMENTARY MATERIAL: LEARNING RELIABLE RULES BY RE-GENERATING DEEP FEATURES

## 1 MORE CASE STUDIES OF MSFS

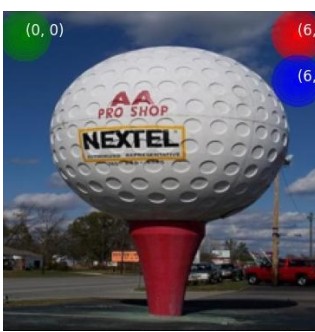 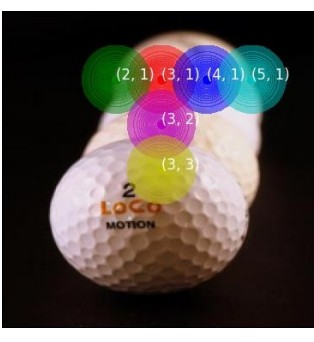 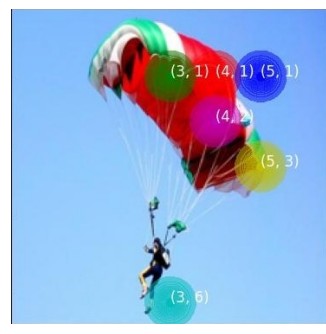

(a) Class "golf ball".    (b) Class "golf ball".    (c) Class "parachute".

Figure 1: **(a)** The MSF of target feature (in red) includes non-neighboring features; **(b)** See analysis of rules in Table 1; **(c)** See analysis of rules in Table 2.