# OpenReview forum: "Learning Reliable Rules by Re-generating Deep Features"
_ICLR.cc/2025/Conference — Submitted to ICLR 2025_

### Official Review · Reviewer_LBBH · 2024-11-02

**Soundness:** 3
**Presentation:** 2
**Contribution:** 2
**Rating:** 5
**Confidence:** 2

**Summary:**

In this paper, the author proposed Visual SATNet (Vi-SATNet) which targets learning logic reasoning rules in a general domain. Specifically, the author trains and evaluates Vi-SATNet on deep feature re-generation and the experiment results show the effectiveness of the proposed methods.

**Strengths:**

1. The experiment results are convincing, showing the effectiveness of the proposed method.
2. The author provides detailed definitions and algorithm processes to demonstrate their ideas and contributions.

**Weaknesses:**

1. The author proposes a method to learn logical reasoning rules in a more general domain. However, different from the original SATNet, which targets solving Sudoku, a task has a clear logic process, how can we understand the logical reasoning for the real-life images? The author should provide more explanation and analysis.
2. In Table 1, the author obtained the best results with regeneration with a 30% mask ratio. However, without regeneration, the model's performance will drop with a higher maks ratio.  Why does the model have the best performance with a 30% mask ratio? The author should provide more analysis.

**Questions:**

N/A

---

> ### Author Response · Authors · 2024-11-21
> **Response to Reviewer LBBH**
>
> We thank the reviewer for the time and effort put into evaluating our work. We address each of the concerns in detail below.
>
> ### Q1: understanding logical reasoning for real-life images.
>
> **A1**: To understand logical reasoning for real life images, we need two critical specifications: 1. a proper abstraction (or representation) over the information encoded in a given image (i.e. the construction of predicates and their meaning), and 2. logical rules on top of a given abstraction.
>
> In this work, we mainly focus on the second point and learns propositional rules on top of the abstraction of feature vectors (since classifiers rely on the feature maps to perform predictions). In fact, it is possible to fit other abstractions into our Vi-SATNet models since they are designed to be a drop-in layer given any abstraction (in this work, feature maps) paired with a reasoner (in this work, a classifier).
>
>
>
> ### Q2: performance peak at 30% masking ratio.
>
> **A2**: We thank the reviewer for the detailed observation in our results. The difference in the performance of the model from 10% masking to 30% masking is marginal. We have evaluated the model again across **ten** runs and the result is summarized in *Table 4* (we have updated these results accordingly in the revised pdf). Increasing the mask ratio makes the reasoning task more difficult and hence results in lower accuracy. We can see that with re-generation of feature vectors given by our model, the classification accuracy maintained above 90% even when 80% of the values are missing, significantly outperforming the vanilla VGG-19.
>
> ### Table 4: Classification accuracy with different blur mask ratio on Imagenette. Mean accuracy and error bar reported on 10 runs for each mask ratio.
>
> | Mask Ratio (%) | w/o Regeneration (%) | w Regeneration (%) |
> |----------|----------|----------|
> | 10  | 97.20$\pm$0.06  | 98.48$\pm$0.01    |
> | 20  | 83.50$\pm$0.09  | 98.49$\pm$0.01    |
> | 30  | 46.20$\pm$0.08  | 98.49$\pm$0.02    |
> | 40  | 23.38$\pm$0.05  | 98.44$\pm$0.03    |
> | 50  | 11.73$\pm$0.05  | 98.32$\pm$0.03    |
> | 60  | 9.85$\pm$0.01   | 98.13$\pm$0.05    |
> | 70  | 9.84$\pm$0.004   | 97.45$\pm$0.05    |
> | 80  | 9.83$\pm$0.001   | 94.23$\pm$0.10    |
> | 90  | 9.84$\pm$0.003   | 72.87$\pm$0.18    |
> | 100 | 9.84$\pm$0.005   | 6.83$\pm$0.11     |

---

### Official Review · Reviewer_HqXe · 2024-11-03

**Soundness:** 2
**Presentation:** 2
**Contribution:** 3
**Rating:** 5
**Confidence:** 3

**Summary:**

This manuscript introduces Visual SATNet (Vi-SATNet), an innovative extension of SATNet aimed at learning logical rules within complex feature spaces, particularly those generated by convolutional neural networks (CNNs). By employing Vi-SATNet for feature regeneration, the study illustrates its capacity to improve interpretability in CNNs during image classification tasks. Experiments conducted on datasets such as MNIST and Imagenette reveal promising classification outcomes across different levels of feature masking.

**Strengths:**

1.	Vi-SATNet extends the capabilities of the original SATNet by enabling the learning of logical reasoning rules within the feature spaces of convolutional neural networks (CNNs). This generalization allows it to operate effectively on complex datasets beyond simple logical puzzles.
2.	The study presents a new method for feature regeneration that leverages learned logical rules derived from deep features, thereby enhancing the interpretability and reliability of convolutional neural networks (CNNs).
3.	Vi-SATNet can be seamlessly integrated as a drop-in layer within existing CNN architectures, requiring no retraining or fine-tuning.

**Weaknesses:**

1.	The manuscript effectively outlines the architecture of Vi-SATNet and establishes an evaluation framework through feature regeneration tasks, employing cosine similarity and Vi-C agreement as measurable metrics for assessing feature quality. However, it could enhance clarity regarding the Vi-SATNet training process, especially concerning hyperparameter selection.
2.	The manuscript describes visualization through minimal significant feature sets (MSFs) and their corresponding receptive fields. However, a more in-depth explanation of how MSFs correlate with specific learned rules would be advantageous. Without clear criteria for evaluating rule quality, the reader may find it challenging to interpret the rules’ significance.
3.	To enhance the study's rigor, it is recommended to include additional datasets, such as CIFAR-10 or other complex real-world datasets.
4.     Some related works are needed to discuss in the manuscript, such as [1-2].

[1] Wei J, Garrette D, Linzen T, et al. Frequency effects on syntactic rule learning in transformers[J]. arXiv preprint arXiv:2109.07020, 2021.
[2] Zhang W, Mo S, Liu X, et al. Learning robust rule representations for abstract reasoning via internal inferences[J]. Advances in Neural Information Processing Systems, 2022, 35: 33550-33562.

**Questions:**

See above.

---

> ### Author Response · Authors · 2024-11-21
> **Response to Reviewer HqXe**
>
> We appreciate the reviewer's constructive feedback and concise summary of our contributions. Please find our detailed responses to each of the comments in the following.
>
> ### Q1: Vi-SATNet training process and hyperparameters.
>
> **A1**: We thank the reviewer for pointing out the confusion on the hyperparameter selection process. In *Table 3* we include explanations for each of the hyperparameters used during training (taking the Imagenette as an example). It is worth noticing that the Vi-SATNet model is not very sensitive to the values of hyperparameters, which is a nice property inherited from SATNet[4]. During training, we did minimum ablation on the hyperparameters and found insignificant difference in performance. Hence, we present the performance of simple models for evaluation. More results on ablation studies can be included in future work to assess the robustness of Vi-SATNet models.
>
> ### Table 3: Vi-SATNet hyperparameters for Imagenette.
>
> | Symbol | Value | Meaning |
> |----------|--------------|------------|
> | H  | 7    | Height of feature map |
> | W  | 7    | Width of feature map |
> | K  | 512    | Dimension of the feature vectors |
> | n  | WxH = 49    | Number of variables |
> | m  | 500    | Number of clauses |
> | mask_ratio  | various values    | Proportion of missing values in the feature map |
>
>
>
>
> ### Q2: Interpretation and significance of the learned rules; explanation of MSFs with respect to the rules.
>
> **A2**: The detailed explanation of the correlation between MSFs and the learned rules can be illustrated by the following examples.
>
> Consider the following example of an image from the "golf ball" class (this image was included in the paper pdf, but you can also find it in the supplementary material, Figure 1b):
>
> There are around *12K* rules extracted from the weight matrix learned for the class "golf ball" in the form of weighted MaxSAT. This set of rules is a discrete representation of what a Vi-SATNet has learned. The rule extraction procedure follows the one presented in [1]. The first few lines of the extracted rules are shown in *Table 1*. Given a target feature vector and its MSFs, we can pinpoint the rules that are related to variables representing the target feature vector and its MSFs, which on average results in a set of less than *100* rules (depending on the size of MSF). The first few lines of the subset for image 1b with f12 as the target feature is shown in *Table 2*.
>
> ### Table 1: Some rules from the extracted weighted MaxSAT rules for class "golf ball".
>
> | Weight | Rule | Meaning |
> |----------|--------------|------------|
> | 6  | (!f14 and f12) or (f14 and !f12)    | f14 != f12 |
> | 5  | (!f12 and f3) or (f12 and !f3)    | f12 != f3 |
> | 9  | (f2 and f3) or (!f2 and !f3)    | f2 = f3 |
> | ... | ... | ... |
> | In total ~12k lines | | |
>
>
>
> ### Table 2: Some rules that are related to the target feature (f12).
>
> | Weight | Rule | Meaning |
> |----------|--------------|------------|
> | 6  | (!f14 and f12) or (f14 and !f12)    | f14 != f12 |
> | 5  | (!f12 and f3) or (f12 and !f3)    | f12 != f3 |
> | 5  | (!f12 and f4) or (f12 and !f4)    | f12 != f4 |
> | 2  | (!f12 and !f19) or (f12 and f19)    | f12 = f19 |
> | ... | ... | ... |
> | In total <100 lines | | |
>
>
> Now, each feature vector can be mapped back to the discrete (boolean) space by randomized rounding [2]. Hence, we can deploy an external SAT solver[3] to compute the boolean value of any missing feature vector, given the learned rules (which constraint the relations between the features) and the MSFs (which can be discretized and plugged into the formula as known values). By solely inputting the boolean values of the MSFs, the solver is able to correctly output the boolean value for the target feature (in this example, the target feature is f12, and the MSFs are $<$f11, f13, f14, f19, f26$>$, the solver assigns f12 the value of 1). The ground truth boolean value of the target feature is obtained by applying randomized rounding to the feature vector itself (in this example, the ground truth of f12 is also 1).
>
> The exact same procedure can be applied to a different example from the "parachute" class (please kindly find this image in the supplementary material, Figure 1c), with target feature being f13, and MSF indices being $<$12, 14, 20, 28, 47$<$. The solved value for f13 is 1 and the ground truth value for f13 is 1 as well.

---

> > ### Author Response · Authors · 2024-11-21
> > **Response to Reviewer HqXe (Part 2)**
> >
> > ### Q3: evaluation on real-world datasets.
> >
> > **A3**: We thank the reviewer for consideration on the dataset complexity. We would like to point out that the Imagenette dataset is a subset of ImageNet, which is often considered as very complex real-world images. Comparing to the CIFAR datasets which have a resolution of 32x32, Imagenette has a resolution of 320x320. Nevertheless, our architecture is designed to be generalizable to any dataset provided with the corresponding CNN model. We can indeed include ablation studies on the generalization ability of Vi-SATNet to different datasets and classification models.
> >
> >
> > ### Q4: Other related works.
> >
> > **A4**: We thank the reviewer for mentioning these recent works. We will add a subsection under the related work section to discuss abstract rule learning on various tasks including general reasoning on images and syntactic rule learning on textual data.
> >
> > [1] Zhaoyu Li, Jinpei Guo, Yuhe Jiang, and Xujie Si. Learning reliable logical rules with satnet. Advances in Neural Information Processing Systems, 36:14837–14847, 2023.
> >
> > [2] Goemans, M. X. and Williamson, D. P. Improved approximation algorithms for maximum cut and satisfiability problems using semidefinite programming. Journal of the ACM (JACM), 42(6):1115–1145, 1995.
> >
> > [3] Zhendong Lei, Shaowei Cai, Dongxu Wang, Yongrong Peng, Fei Geng, Dongdong Wan, Yiping Deng, and Pinyan Lu. Cashwmaxsat: Solver description. MaxSAT Evaluation, 2021.
> >
> > [4] Po-Wei Wang, Priya Donti, Bryan Wilder, and Zico Kolter. SATNet: Bridging deep learning and
> > logical reasoning using a differentiable satisfiability solver. In International Conference on Machine Learning, 2019.

---

### Official Review · Reviewer_CKFZ · 2024-11-04

**Soundness:** 3
**Presentation:** 2
**Contribution:** 2
**Rating:** 6
**Confidence:** 3

**Summary:**

This paper extends SATNet by enabling it to learn logical rules from the complex feature space of real-life images, allowing it to regenerate masked features while maintaining high classification accuracy.
Experimental results on the features from some pre-trained models show that the learned reasoning rules allow Vi-SATNet to re-generate missing feature vectors accurately.
Finally, the authors present a visualization technique to illustrate the rules learned by Vi-SATNet models.

**Strengths:**

Experiments on MNIST and Imagenette datasets demonstrate the effectiveness of Vi-SATNet in feature regeneration and classification, showing impressive results even under high masking ratios.

The visualizations of the learned rules offer an intuitive understanding of feature dependencies.

**Weaknesses:**

The visualization of the learned rules is not very clear. Whether in the foreground or background, MSF tends to pay more attention to the position around the target, which is a rather trivial result.

**Questions:**

As mentioned in the weaknesses section, the visualization of the learned rules indicates that the target-related MSF pays more attention to its surrounding features, which is a trivial finding. Since CNN features are derived from convolutions with neighboring pixels, the features at each position are inherently closely related to those in the surrounding area. Could you provide some more meaningful rules from different perspectives?

The features in the paper appear to be derived from the last layer of the convolutional module in the model. What would happen if features from other convolutional layers were used? Could different layers with varying depths extract distinct rules?

---

> ### Author Response · Authors · 2024-11-21
> **Response to Reviewer CKFZ (Part 1)**
>
> We thank the reviewer for the nice summary and the thoughtful questions. Below, we address the comments point-by-point.
>
> ### Q1.1: MSFs are surrounding features.
>
> **A1.1**: We completely agree with the reviewer that CNN features are inherently closely related to the surrounding features. However, we would like to point out that *not* all surrounding features are included in the minimal significant feature set (MSFs) found. Specifically, consider the four direct neighbors of a target feature (up, down, left, right): in the MSFs shown in the results, we can see a clear *inclusion* of neighboring features that are in the foreground (if target feature is in the foreground) and *exclusion* of neighboring features that are in the background. This observation precisely shows the meaning of an MSF: only the features that are significant to the generation of the target feature are identified.
>
> Please kindly find a new example in the supplementary material (Figure 1a), which has an MSF that includes a *top left* feature located at position (0,0) for the target feature that is in the *top right* corner located at position (6,0). The two features included in the MSF are both background features (since the target feature is also in the background) and one of them is *not* a neighboring feature of the target feature.
>
> We hope that this example clarifies the concern of MSFs only containing neighboring features.
>
> ### Q1.2: Rule interpretation from other perspectives.
>
> **A1.2**: We thank the reviewer for asking this question. As mentioned in the paper, we chose to present the learned rules via visualization of the receptive fields of MSFs because it is the most straight forward interpretation method. A different method to present the learned rules is by directly showing the extracted propositional formulas on the boolean variables, where each variable represents an abstracted state of the corresponding feature vectors. Consider the following example of an image from the "golf ball" class (this image was included in the paper pdf, but you can also find it in the supplementary material, Figure 1b):
>
> There are around *12K* rules extracted from the weight matrix learned for the class "golf ball" in the form of weighted MaxSAT. This set of rules is a discrete representation of what a Vi-SATNet has learned. The rule extraction procedure follows the one presented in [1]. The first few lines of the extracted rules are shown in *Table 1*. Given a target feature vector and its MSFs, we can pinpoint the rules that are related to variables representing the target feature vector and its MSFs, which on average results in a set of less than *100* rules (depending on the size of MSF). The first few lines of the subset for image 1b with f12 as the target feature is shown in *Table 2*.
>
> ### Table 1: Some rules from the extracted weighted MaxSAT rules for class "golf ball".
>
> | Weight | Rule | Meaning |
> |----------|--------------|------------|
> | 6  | (!f14 and f12) or (f14 and !f12)    | f14 != f12 |
> | 5  | (!f12 and f3) or (f12 and !f3)    | f12 != f3 |
> | 9  | (f2 and f3) or (!f2 and !f3)    | f2 = f3 |
> | ... | ... | ... |
> | In total ~12k lines | | |
>
>
>
> ### Table 2: Some rules that are related to the target feature (f12).
>
> | Weight | Rule | Meaning |
> |----------|--------------|------------|
> | 6  | (!f14 and f12) or (f14 and !f12)    | f14 != f12 |
> | 5  | (!f12 and f3) or (f12 and !f3)    | f12 != f3 |
> | 5  | (!f12 and f4) or (f12 and !f4)    | f12 != f4 |
> | 2  | (!f12 and !f19) or (f12 and f19)    | f12 = f19 |
> | ... | ... | ... |
> | In total <100 lines | | |
>
>
> Now, each feature vector can be mapped back to the discrete (boolean) space by randomized rounding[2]. Hence, we can deploy an external SAT solver[3] to compute the boolean value of any missing feature vector, given the learned rules (which constraint the relations between the features) and the MSFs (which can be discretized and plugged into the formula as known values). By solely inputting the boolean values of the MSFs, the solver is able to correctly output the boolean value for the target feature (in this example, the target feature is f12, and the MSFs are $<$f11, f13, f14, f19, f26$>$, the solver assigns f12 the value of 1). The ground truth boolean value of the target feature is obtained by applying randomized rounding to the feature vector itself (in this example, the ground truth of f12 is also 1).

---

> > ### Author Response · Authors · 2024-11-21
> > **Response to Reviewer CKFZ (Part 2)**
> >
> > ### Q2: Input feature maps from other layers in an CNN.
> >
> > **A2**: In this work, we selected the feature maps from the last convolution layers because they are representations at the highest level of abstraction, which should yield most interpretable logical rules. Using feature maps from other convolution layers as the input to our model is definitely possible, and distinct rule sets are anticipated to be learned since the input space is at a different level of abstraction (i.e. the meaning of the boolean variables in the learned rules would be different at each layer). In other words, low-level features would yield a set of low-level reasoning rules, for example, propositional rules that directly reasons on the pixel space, which would be more local and less interpretable.
> >
> > [1] Zhaoyu Li, Jinpei Guo, Yuhe Jiang, and Xujie Si. Learning reliable logical rules with satnet. Advances in Neural Information Processing Systems, 36:14837–14847, 2023.
> >
> > [2] Goemans, M. X. and Williamson, D. P. Improved approximation algorithms for maximum cut and satisfiability problems using semidefinite programming. Journal of the ACM (JACM), 42(6):1115–1145, 1995.
> >
> > [3] Zhendong Lei, Shaowei Cai, Dongxu Wang, Yongrong Peng, Fei Geng, Dongdong Wan, Yiping Deng, and Pinyan Lu. Cashwmaxsat: Solver description. MaxSAT Evaluation, 2021.

---

> > > ### Comment · Reviewer_CKFZ · 2024-11-27
> > >
> > > Thanks to the authors for their efforts and responses, which have addressed part of my concerns. Hence, I will increase my score to a 6.

---

### Author Response · Authors · 2024-11-25
**Dear reviewers, please kindly let us know if you have further questions, concerns or suggestions**

Dear reviewers, please kindly let us know if you have further questions, concerns or suggestions regarding to our work, and we would be very happy to answer them.

---

### Meta-Review · Area_Chair_RyR1 · 2024-12-19

**Metareview:**

The proposed Vi-SATNet extends the capabilities of the original SATNet by enabling the learning of logical reasoning rules within the feature spaces of convolutional neural networks (CNNs). The study presents a new method for feature regeneration that leverages learned logical rules derived from deep features, thereby enhancing the interpretability and reliability of convolutional neural networks (CNNs).However, the explaniation of the logical reasoning for real-life image is still unclear. And the experimental results as well as the analysis are limited.

**Additional Comments On Reviewer Discussion:**

Dear reviewers,
Thanks a lot for the reviewing work. Have a good holiday!

---

### Decision · Program_Chairs · 2025-01-22

Reject